# Nutritional Aspects of Commercially Available Complementary Foods in New Zealand Supermarkets

**DOI:** 10.3390/nu12102980

**Published:** 2020-09-29

**Authors:** Shanjivan Padarath, Sarah Gerritsen, Sally Mackay

**Affiliations:** 1Department of Primary Care and Nutrition, School of Public Health and Primary Care, Fiji National University, Suva, Fiji; spad267@aucklanduni.ac.nz; 2Department of Social and Community Health, School of Population Health, University of Auckland, Auckland 1023, New Zealand; s.gerritsen@auckland.ac.nz; 3Department of Epidemiology and Biostatistics, School of Population Health, University of Auckland, Auckland 1023, New Zealand

**Keywords:** complementary feeding, commercial complementary food, baby food, infant, toddler, sugar, sodium, food allergy, texture, flavour

## Abstract

Optimal nutrition in early childhood fosters growth and development whilst preventing morbidity and mortality in later life. There is little research in New Zealand on commercially available complementary foods (CACFs). This cross-sectional study of the nutritional aspects and packaging of CACFs used data collected in four major supermarket chains in New Zealand in 2019 (Nutritrack). Of the 197 CACFs analysed, 43 (21.8%) were inappropriately recommended for consumption by children four months of age or older, 10 (5.1%) had added salt, and 67 (34.0%) contained free sugars. The majority (*n* = 136, 69.0%) contained ingredients with a sweet flavour. Relatively sweet vegetables like carrot and sweetcorn were used more often than bitter vegetables such as broccoli and spinach. The described texture of most (*n* = 145, 62.1%) wet ‘spoonable’ products was of the lowest complexity (smooth, puréed, custard). CACFs would adequately expose children to cow’s milk and wheat but not to other common food allergens (cooked hen’s egg, soy, fish, crustacean shellfish, peanut, and tree-nuts). If children’s diets include CACFs, non-commercial meals must be offered as well in order to meet nutritional guidelines related to the introduction of common food allergens, diversity of flavours, and complex textures for infants and toddlers.

## 1. Introduction

Nutrition in the first 1000 days of life is critical for growth and development, lays the foundation for eating behaviour, determines food preferences, and influences health outcomes in later life [1,2,3,4,5,6]. These 1000 days encompass complementary feeding, a bridging period from six to twenty-four months of age when children learn to adapt to family food via the gradual introduction of solids and liquids alongside their usual milk (breastmilk and or substitutes) [1,2,3,4].

The foods chosen for children by care providers are dependent on complex social, economic, cultural, and political determinants of the food environment [3,4,5,6,7]. As such, a carer may choose to feed their child commercially available complementary foods (CACFs). This choice must be supported by all infant and young child feeding stakeholders [3]. Since their ‘invention’ about a hundred years ago in North America, the use of CACFs has become commonplace in households across the world due to their long shelf-life, portability, convenience, relatively low cost, and assumed nutritional value [8]. The Growing Up in New Zealand study found that CACFs like baby rice, baby breakfast cereals, and rusks were some of the first complementary foods introduced into the diets of infants at around five to six months of age [9]. By nine months of age, baby breakfast cereals were a staple in the diets of New Zealand’s youngest, whereby 47.9% of children consumed these daily and a further 24.5% consumed them weekly [9]. The use of CACFs as a source of nourishment continued into toddlerhood, with 60.0% of children aged 12–24 months having had consumed at least one CACF daily [10]. The commercial baby food sector in New Zealand grew by 27.2% from 2010 to 2016 and had a retail value of $42.6 million in 2019 [11,12].

Dietary guidelines and legislature on CACFs are essential to ensure that infants and young toddlers consume foods which are safe, timely, adequate, and appropriate. The *Global Strategy on Infant and Young Child Feeding* encourages World Health Organization (WHO) member states to create national policies and strategies for infant feeding [3]. In New Zealand, the fourth edition of the Ministry of Health’s *Food and Nutrition Guidelines for Healthy Infants and Toddlers (0–2)* currently provides dietary recommendation on CACFs [4]. The *Australia New Zealand Food Standards Code—Standards 2.9.2—Food for Infants*, which is based off the Codex Alimentarius is the current legislature that regulates CACFs for infants under 12 months old in New Zealand [13]. These two guiding documents recommend the following:Complementary feeding should be initiated at around six months of age when developmentally ready.Prepare or choose pre-prepared foods with no added fat, salt, sugar, honey, or other sweeteners.Upper allowable limits for sodium in CACFs recommended for infants less than 12 months old are as follows: rusks—350 mg/100 g, biscuits—300 mg/100 g, and all other CACFs—100 mg/100 g.The texture, variety, flavour, and amount of foods should increase as the infant progresses through the developmental stages to join the family diet from 12 months of age.Infant foods containing common food allergens like milk and milk products can be introduced from six months of age.

The WHO recommends that complementary feeding be introduced at six months (180 days) of age alongside breastfeeding, which is to continue for up to two years or beyond [1]. It is recommended that the first solid foods are vegetables to prevent reinforcing innate preference for sweet foods and to increase the acceptability of vegetables [6,14,15]. Fruits, sweet and bitter vegetables, and proteins of diverse flavours and textures should be introduced gradually and repeatedly into the diet of children prior to nine months of age in order to foster acceptance of diverse foods, prevent feeding difficulties, and augment oral development [1,4,6,14,16,17,18]. Complementary foods need to be devoid of salt and sugar to prevent the development of preferences for these foods, as chronic and excessive intake is associated with the increased risk of metabolic diseases in later life [1,4,6]. During the first year of life, infants need to be introduced to common food allergens like cow’s milk, soy, fish, shellfish, cooked hen’s egg, wheat, peanut, and tree-nuts as a means of reducing the risk of atopy and allergies in early and late childhood [19,20].

The WHO Guidance on Ending the Inappropriate Promotion of Foods for Infants and Young Children and the International Code of Marketing of Breast-milk Substitutes state that the marketing of CACFs and milk substitutes for infants less than six months old should be prohibited [21,22]. However, analyses of the European market in 2019 showed that 27.9% of CACFs in Israel, 44.0% of CACFs in Austria, 53.0% of CACFs in Hungary, and 60.0% of CACFs in Bulgaria were inappropriately marketed for children under six months old [23]. In The United Kingdom (UK), food market surveys showed that whilst there was a significant reduction in the percentage of products marketed for children less than six months old, from 43.0% in 2013 to 23.0% in 2019, the number of discrete products targeted for the same age-group actually rose from 178 to 201 different product types [24].

Introducing complex textures into the diet of infants augments oral development and prevents the development of eating behaviours later in life [14,25,26]. Yet, CACFs found in the UK for infants have been noted to be mostly ‘smooth’ or ‘smooth with lumps’ which would not encourage a faster progression to more complex textures (‘mashed’ and ‘chopped’) [27]. Similarly, analysis of CACF in the United State of America (USA) by Beauregard et al. (2019) found that of the 1073 products available for purchase, 67.8% (*n* = 703) were described to have a ‘puréed’ consistency that did not require chewing [28]. The First Steps Nutrition Trust Survey in 2017 found that compared to CACFs, the same recipe when home-made had a more complex texture because more vegetables and protein were used with minimal to no water. Conversely, CACFs were more likely to have water added during industrial processing and then were re-textured using thickeners such as flours and starches [27].

Analysis of CACFs in USA and Europe has shown that most foods are low in sodium and comply with national dietary guidelines [23,29]. Likewise, salt was rarely added into these commercial foods directly. However, ingredients which are deemed to be salty such as cheese, processed meats like ham, and stock were present in main meals [23,27].

The general global consensus on free sugars is that they should not be added to foods for children less than two years of age [1,4,5,6]. Almost 30.0% of CACFs in Austria, 32.6% of CACFs in Israel, 37.5% of CACFs in Hungary, and 41.4% in Bulgaria contained added sugars and sweeteners like fruit juices and concentrates [23]. In USA, more than 70.0% of CACFs for toddlers contained one or more added sugars [29].

Timely and repeated introduction of diverse flavours one at a time, especially those imparted by bitter cruciferous vegetables encourage acceptance of these flavours and increases consumption in childhood and later life [15,30,31]. However, analyses of CACFs found in USA and the UK have found that most products were made up of multiple fruits and vegetables that would not allow consumers to discern between distinct flavours. They also had an overall sweet flavour due to the preferential use of relatively sweet vegetables and fruit juice concentrates [27,28,29,32].

Therefore, considering that CACFs in Europe and USA are diverging from dietary guidelines and regulations, it was imperative that CACFs in New Zealand be analysed too. The objectives of this study were to describe the CACFs retailing in New Zealand supermarkets in 2019 and determine whether they met national nutritional guidelines regarding recommended age of introduction from six months, increasing texture with age, the addition of salt, sugar, and common food allergens, and variety of flavour and fruit/vegetable content. Additionally, the allowable limits for sodium content were checked for compliance to the Codex derived national regulations.

## 2. Materials and Methods

Data were extracted from the online database, Nutritrack, which is maintained by the National Institute of Health Innovation at the University of Auckland [33]. The nutritional information on commercially available foods and beverages are collected by trained interviewers from four major supermarkets using a smartphone application, annually. Photographs are taken of all aspects of the packaging, including ingredient lists and Nutrition Information Panels. The database represented 75.0% of products on the New Zealand market and in 2019 had an accuracy of 99.1%, as determined by stringent quality control audits [34].

A CACF was defined as a commercial product marketed for complementary feeding by children less than 24 months of age, as determined by the product label and/or directions of use. CACFs could either be wet ‘spoonable’ or dry products like cereals and snacks. One product was excluded, a multi-portioned box of biscuits, because it did not have a recommended age. Products that were of different flavours and those which were packaged in different forms were considered as discrete CACFs because they were noted to have different formulations. Once a CACF was identified, it was classed into four mutually exclusive food categories (breakfasts, main meals, desserts, and snacks) based on product name and indication of use. Breakfast meals were those products that were single-portioned ready-to-eat cereal meals and multi-portioned dry cereals meant to be reconstituted with baby’s usual milk or water. The main meals category was further sub-divided into the following meal sub-groups as determined by the main ingredient and the product name: meat-based, poultry-based, fish-based, fruit-based, fruit and vegetable-based, and vegetable-based. The desserts category contained products marketed as puddings and custards. The final category was that of snacks which were dry single or multi-portioned sweet products and rusks. The pictures of products on Nutritrack helped determine whether the packaging was that of squeezable pouches, bags, boxes, cans, glass jars, or plastic containers. A squeezable-pouch is an aluminium-lined squeezable packaging that has a twist-off re-sealable plastic cap.

The products were classified into five stages, based on the manufacturers’ recommended consumer age-group. Stage 1 (4 months or older), Stage 2 (6 months or older), Stage 3 (8 months or older), Stage 4 (10 months or older), and Stage 5 (12 month or older).

Information on the texture of wet ‘spoonable’ CACFs was found on the product label. These described textures were tabulated along an age gradient from Stage 1 to Stage 5, to determine whether there was progression in texture. The Food and Nutrition Guidelines recommend that the texture of food must increase from ‘puréed’ to ‘mashed’ to ‘chopped’.

The Nutrient Information Panel (NIP) contained the sodium content of the products. These were then compared to the reference ranges detailed in the *Australia New Zealand Food Standards Code—Standard 2.9.2*, which is the legislative document for CACFs recommended for infants under the age of 12 months old in New Zealand. An upper limit for allowable sodium content was identified as 350 mg/100 g of sodium for rusks, 300 mg/100 g of sodium for biscuits, and 100 mg/100 g of sodium for other CACFs marked for infants [13]. There is no regulation on nutrient content of CACFs marketed for children older than 12 months and so these products were not included in the sodium analyses. Vegetable and protein-based stock were regarded as an ‘added salt’ since they are not a main nutritional feature of meals, rather are added to enhance the flavour profile. The ingredient list was scrutinized to determine the presence of salt, free sugars, and common food allergens. Free sugars were defined as all mono-saccharides and di-saccharides added to foods by the manufacturer during production, including sugars that are naturally present in honey, fruit juices, fruit juice concentrates, and syrups [35]. The eight common food allergens included in this study were: cow’s milk and its derivatives (cheese and yoghurt), soy, peanut, tree-nuts, fish, crustacean shellfish, gluten (wheat, barley, oats, and rye), and hen’s egg [19,36].

The overall flavour of CACFs was determined by categorising the ingredients they contained into sweet and savoury flavours using the methods described by Garcia et al. (2016) [32]. Breakfast meals, snacks, and desserts that contained free sugars or fruits were classified as sweet. Fruit-based meals were classified as sweet. Fruit and vegetable-based meals in which the majority (>50.0%) of ingredients were fruits were classified as sweet. Vegetable-based meals which had a majority (>50.0%) of relatively sweet vegetables and no bitter vegetables were classified as sweet. Fish-based, meat-based, and poultry-based meals were classified as savoury even if free sugars were added. The *New Zealand Food Composition Database* was used to determine the total sugar content of vegetables. Any vegetable that contained more than 4 g/100 g of total sugar was classified as being relatively sweet [37,38]. Cruciferous vegetables like kale, broccoli, and cauliflower along with spinach were classified in this study as bitter vegetables [39].

Descriptive analyses on the addition of free sugars, salt, types of fruits and vegetables, and common food allergens, the sodium content, packaging, texture, flavours, and recommended age of intended consumer were conducted using Microsoft Excel.

Approval to access Nutritrack was obtained from the National Institute of Health Innovation, University of Auckland.

## 3. Results

### 3.1. Description of CACFs

In 2019, there were 197 discrete CACFs in New Zealand supermarkets, retailing under twelve different brand names. There were 153 (77.7%) wet ‘spoonable’ CACFs and 44 dry products. The most abundant category of CACFs was main meals (*n* = 110, 55.8%) (Table 1). All fish-based (*n* = 1), meat-based (*n* = 30), poultry-based (*n* = 14), fruit-vegetable-based (*n* = 13), and two thirds of vegetable-based meals (*n* = 13, 68.4%) were composite in nature such that the meals consisted of ingredients from different food groups, for example, vegetable risottos and meat pasta dishes. All of the vegetable-based meals contained two or more vegetables as ingredients. Most fruit-based CACFs (*n* = 28, 84.8%) were composed of two or more fruits whilst the remaining six fruit-based products contained only one ingredient. Eleven products (5.6%) required reconstitution prior to consumption while the remaining 186 were either single or multi-portioned ready-to-eat CACFs. More than half of the CACFs were packaged in squeeze pouches (*n* = 103, 52.3%), and the rest were packed into bags (*n* = 47, 23.9%), cans (*n* = 17, 8.6%), glass jars (*n* = 15, 7.6%), boxes (*n* = 13, 6.6%), and plastic containers (*n* = 2, 1.0%). Six products (3.0%) recommended for Stage 5 [12m+] children were packaged in squeezable pouches.

### 3.2. Recommended Age for Consumption

The majority (*n* = 161, 81.7%) of all CACFs were recommended for children less than 12 months of age [Stages 1–4]. As shown in Table 1, 21.8% of CACFs were meant to be consumed by Stage 1 [4m+] children, 35.0% for Stage 2 [6m+] children, 18.8% for Stage 3 [8m+] children, 6.1% for Stage 4 [10m+] children, and 18.3% for Stage 5 [12m+] children.

### 3.3. Texture

Of the 153 wet ‘spoonable’ products, 145 had their texture described by the manufacturer. The majority (n = 90, 62.1%) of these CACFs were of the lowest textural complexity (smooth, puréed, super smooth, and custard). Analysis as per product description showed that textures became more complex along the age-gradient for all vegetable-based, meat-based, and poultry-based main meals (Table 2). However, a clear progression in complexity of textures along the age-gradient was not evident for ready-to-eat breakfasts and fruit-based meals.

### 3.4. Added Salt and Sodium Content

Ten (5.1%) CACFs were found to contain an added salt and these were from the following food sub-groups: two each of breakfast cereals, meat-based meals, and vegetable-based meals; and four rusks. Five (2.5%) of the CACFs were recommended for Stage 5 children aged more than 12 months old and five were for children six to twelve months old (Stages 2–4). The amount of sodium in all products recommended for children less than 12 months of age was below the regulatory thresholds.

### 3.5. Added Free Sugars

More than a third (*n* = 67, 34.0%) of CACFs available for purchase contained free sugars. These were in the form of sugars (raw, cane, grape), maltodextrin, fruit juices (apple, grape, pineapple, purple carrot, cherry, strawberry, passionfruit, and red beet), dextrose, fruit juice concentrates (apple, pear, grape, carrot, purple carrot, and elderberry), and glucose.

As demonstrated in Table 3, free sugars were present in 100.0% of desserts, 55.2% of sweet snacks, 36.4% of ready-to-eat breakfast meals, and 33.3% of fruit-based meals.

Free sugars were added in 47.2% of Stage 5 [12m+] CACFs, 34.8% of Stage 2 [6m+] products, 32.4% of Stage 3 [8m+] products, 23.8% of Stage 1 [4m+] products, and 16.7% of Stage 4 [10m+] products.

### 3.6. Common Food Allergens

Cow’s milk was used in 67 (34.0%) of the CACFs in the form of whole milk, yoghurt, and cheeses. Wheat was present in almost a quarter of CACFs (*n* = 47, 23.9%), mostly in the form of cereals in breakfast meals and flour in main meals. Hen’s egg was present in four (2.0%) main meals as a constituent of pasta. Soy was present as flour in four (2.0%) products. Fish was present in only one (0.5%) product. Peanut, tree-nuts, and crustacean shellfish were absent as ingredients in all CACFs available for purchase in New Zealand supermarkets.

### 3.7. Vareity of Flavour and Fruit and Vegetable Content

There were 175 (88.8%) CACFs that contained at least one fruit and or vegetable. Of these, 93 (47.2%) contained only fruits, 61 (31.0%) contained only vegetables, and 21 (10.7%) contained both fruits and vegetables. There were 21 discrete types of fruits present in CACFs of which the most commonly used were apples (*n* = 53, 26.9%), followed by bananas (*n* = 36, 18.3%), pears (*n* = 20, 10.2%), mangos (*n* = 19, 9.6%), and blueberries (*n* = 14, 7.1%). There were 24 discrete types of vegetables used in CACFs, of which carrots (*n* = 54, 27.4%), onions (*n* = 43, 21.8%), sweet potatoes (*n* = 31, 15.7%), tomatoes (*n* = 31, 15.7%), and pumpkin (*n* = 29, 14.7%) were used more frequently. Relatively sweet vegetables like carrots, sweet potatoes, and sweetcorn were used more predominantly in CACFs than bitter vegetables such as spinach, broccoli, and cauliflower (Table 4).

Sixty-nine percent (*n* = 136) of CACFs would impart an overall sweet flavour as they contained sweet flavoured ingredients. All ready-to-eat breakfast meals (*n* = 22), desserts (*n* = 21), fruit-based meals (*n* = 33), and sweet snacks (*n* = 29) contained sweet ingredients. Sixty percent of rusks (*n* = 3) and dry cereals (*n* = 6), 63.2% of vegetable-based meals, and 76.9% of fruit and vegetable-based meals contained sweet ingredients. All of the fish-based (*n* = 1), meat-based (*n* = 30), and poultry-based (*n* = 14) meals had ingredients with savoury flavours. There were 17 products (8.7%) that contained bitter vegetables and they may impart an overall bitter flavour in the product. However, these same 17 products contained one or more sweet vegetables as an ingredient too.

## 4. Discussion

CACFs which meet dietary guidelines and regulations are deemed to be appropriate sources of nourishment for infants and young children. Our research, however, shows that not all products are fully compliant. More than a fifth (21.8%) of products were marketed for children less than six months old, the texture of fruit-based meals and ready-to-eat breakfast meals did not become more complex along the age gradient, and more than a third (34.0%) of products contained free sugars. Children mostly fed CACFs would neither be able to appreciate distinct and diverse flavours nor have adequate exposure to common food allergens like cooked hen’s egg, peanut, tree nuts, fish, crustacean shellfish, and soy. Despite national recommendations suggesting that complementary feeding be initiated around six months of age, more than a fifth of CACFs were still recommended for children four months of age and older. The labelling of products with ‘4 months’, ‘from 4 months’, or ‘from 4 to 6 months’ has been shown to encourage Australian and New Zealand parents to initiate complementary feeding closer to four months rather than closer to six months [40]. Early introduction of complementary feeding has the propensity to displace breastmilk and breast-milk substitutes, which can result in malnourishment if the complementary food is nutritionally inadequate and high in salts and free sugars which are not required in early childhood [1,4,5,6,41,42]. The presence of commercial foods on the market for children younger than six months of age implies that these foods are nutritionally adequate substitutes to baby’s usual milk, which they are not [1,4,42]. Although the percentage of CACFs recommended for children four months of age and older in New Zealand (21.8%) is much lower than in many other European and North American countries, the mere presence of these products are in violation of the *International Code of Marketing of Breast-milk Substitutes and the Guidance on Ending Inappropriate Promotion of Foods for Infants and Young Children* [21,22,23,27,28,37]. Endorsement by healthcare providers for Stage 1 [4m+] products contradict the national *Food and Nutrition Guidelines* [3,43]. The largest provider of health-related support services for children under five in New Zealand has had a 30-year relationship with a CACF manufacturer, resulting in their organization’s logo being promoted on 35.0% (*n* = 15) of Stage 1 [4m+] CACFs [44]. Further, the persistence of products recommended for children less than six months old may be due to the fact that the national Food Standards Code 2.9.2 contain regulations on how cereals for infants aged four months old can be formulated. This indicates to manufacturers that these products can be produced for the market [13].

The most prevalent form of packaging in New Zealand was the squeezable pouch (*n* = 103, 52.3%). Most (*n* = 95, 92.2%) CACFs packaged in squeezable pouches had little textural complexity and were described as puréed, creamy smooth, smooth, or mashed. Such CACFs can be suckled directly from the packages, impeding oral development that is usually stimulated from chewing complex textures [45,46,47]. Children fed commercial meat-based, poultry-based, and vegetable-based main meals appropriate for their Stages 1–5 would be able to experience complex textures as they age. Increase in textural complexity of foods, from smooth and puréed to lumpy or chunky, if introduced prior to nine months of age, enables children to accept novel textures when challenged later in life and augments mastication and oral development [6,26,48,49].

A small proportion of CACFs contained added salt (*n* = 10, 5.1%) which was not in line with the *Food and Nutrition Guidelines*. Our results were congruent with findings of studies conducted in the United Kingdom, the United States, and Canada, which have found that only a few CACFs in those markets contained added salts [23,27,28]. It should be noted that half of the CACFs that had an excess of sodium in our study did not have salt as an ingredient but contained salt indirectly via the use of ingredients like cheeses and processed meats. All CACFs recommended for consumption by children less than 12 months of age had appropriate levels of sodium content.

A significant proportion (*n* = 67, 34.0%) of CACFs contained free sugars despite the *Food and Nutrition Guidelines* recommending that sugar, honey and other sweeteners not be added to complementary foods [4]. Children who are regularly fed CACFs would repeatedly be exposed to sweetened foods which would foster their innate preference for sweet flavours over salty, sour, and bitter flavours [15,30,50,51]. Chronic exposure to foods with free sugars increases the risk of developing dental caries and metabolic diseases in childhood and in later life [1,4,52].

This analysis showed that 69.0% (*n* = 136) of CACFs contained ingredients that would impart an overall sweet flavour. While 17 (8.7%) different products did contain a bitter vegetable, they also had sweet vegetables like carrots, parsnip, sweetcorn, sweet potatoes, and pumpkin as ingredients. The bitterness in foods can be damped by the addition of sweet and salty flavours [53,54]. Manufacturers purposefully combine fruit-based ingredients and sweet vegetables to mask bitter flavours imparted by cruciferous vegetables and spinach; to make such CACFs more palatable [27]. While this strategy may increase the ingestion of bitter vegetables, it neither increases their acceptability nor allows for learning of flavours [16,17]. The New Zealand *Food and Nutrition Guidelines* recommend that children should be introduced to diverse flavours one at a time and repeatedly to enable learning and acceptance [4]. With the exception of five fruit-based meals, 97.5% of CACFs in New Zealand would not allow for broad learning of flavours as they contained two or more ingredients simultaneously. These findings are not novel, CACFs in the United Kingdom and Germany were found to contain disproportionately more relatively sweet vegetables like parsnip, carrots, onions, pumpkin, sweet potatoes, and tomatoes as compared to bitter vegetables like broccoli, cauliflower, kale, chard, and cabbages [32,37]. Garcia et al. (2016) found in their study that 18.0% of CACFs in the United Kingdom contained fruit juices and concentrates, most likely as a sweetener [32]. A similar proportion (*n* = 36, 18.3%) was noted in our study too. Manufacturers should provide more CACFs that would enable learning of bitter flavours by reducing reliance on fruit-based ingredients and sweet vegetables.

The risk of developing food-induced allergies to common food allergens is reduced when these foods are introduced to the general low-risk population in their first year of life as compared to when delayed [19,20]. Since more than a third of products (*n* = 67, 34.0%) contained cow’s milk and almost a quarter (*n* = 47, 23.9%) contained wheat, children fed a variety of CACFs would be adequately exposed to these two allergens in their first year of life. However, children would need to be fed non-commercial meals containing soy, cooked hen’s egg, fish, crustacean shellfish, peanut, and other tree nuts as CACFs are unlikely to expose them to these food allergens.

The major strength of this cross-sectional study is the use of data collected by means of robust and reliable methods which represented 75.0% of the commercial foods available for purchase. A limitation was that CACFs marketed in specialty shops, ethnic outlets, and online were not included, although this is likely to be a very small proportion of total CACFs available on the market. Secondly, the nutritional information panel was utilized to extract the macronutrient content rather than the chemical analysis of the product itself. Actual values of nutrients may vary as manufacturers calculate the nutrient content from individual raw ingredients [27]. However, using descriptors off labels is indicative of real-life situations where consumers purchase products based on the information present on labels only.

## 5. Conclusions

Although CACFs are not the only source of nourishment for children, they do comprise a significant proportion of the diet of children aged six to twenty-four months old in New Zealand. This first descriptive analysis of CACFs in New Zealand has found that most products meet the current recommendations and regulations for salt and sodium but were largely sweet (due to free sugars and relatively sweet ingredients), lacked variety in the vegetables included, and did not contain common food allergens. A fifth of CACFs (21.8%) may encourage the displacement of nutritionally adequate breastmilk and substitutes as they were recommended as appropriate for children from four months of age. Children fed CACFs will also need to be fed nutritionally adequate homemade foods in order to introduce common food allergens and diversity of flavours. Our findings will have implications for all stakeholders. Practitioners must be mindful of the advice they provide regarding complementary feeding. Healthcare organizations must ensure that their endorsements do not contradict national guidelines and regulations. Carers must include non-commercial sources of complementary foods in the diets of children. The authors recommend that dietary guidelines and regulations be updated to reflect current global recommendations so that children are provided with the most appropriate, adequate, acceptable, and safe complementary foods.

## Figures and Tables

**Table 1 nutrients-12-02980-t001:** Count of commercially available complementary foods (CACFs) in New Zealand supermarkets in 2019, stratified by food categories and sub-groups (*n* = 197).

Food Categories and Sub-Groups	Stage 1 [4 m+]	Stage 2 [6 m+]	Stage 3 [8 m+]	Stage 4 [10 m+]	Stage 5 [12 m+]	Grand Total
**Breakfast**	**5**	**15**	**6**		**6**	**32**
Dry cereal	4	5	1			10
Ready-to-eat	1	10	5		6	22
**Desserts**		**15**	**6**			**21**
Custard & Pudding		15	6			21
**Meals**	**38**	**32**	**20**	**8**	**12**	**110**
Fish			1			1
Fruit	22	8	3			33
Fruit & Vege	9	4				13
Meat		13	8	3	6	30
Poultry		6	2	3	3	14
Vegetables	7	1	6	2	3	19
**Snacks**		**7**	**5**	**4**	**18**	**34**
Rusk		2	1		2	5
Sweet		5	4	4	16	29
**Grand Total**	**43**	**69**	**37**	**12**	**36**	**197**

**Table 2 nutrients-12-02980-t002:** Described textures of selected wet ‘spoonable’ commercially available complementary foods (CACFs) in New Zealand supermarkets, stratified by age groups [Stages 1–5] (*n* = 145).

	Stage 1 (4 m+)	Stage 2 (6 m+)	Stage 3 (8 m+)	Stage 4 (10 m+)	Stage 5 (12 m+)
Fruit	Puréed				
	Smooth				
	Super smooth				
Vegetables	Puréed				
	Super smooth				
	Smooth				
		Mashed			
			Soft lumps		
			Fork mashed		
				Lumpy	
					Chunky
Meat & Poultry		Puréed			
		Mashed			
		Fork Mashed			
			Soft lumps		
				Lumpy	
			Chunky		
Ready-to-eat cereal	Super smooth				
			Creamy smooth		
		Puréed			
		Smooth			
		Mashed			
		Fork mashed			
			Soft lumps		

Shading has been used to show that CACFs from different stages along the age-gradient have the same described texture.

**Table 3 nutrients-12-02980-t003:** Count of commercially available complementary foods (CACFs) in New Zealand supermarkets in 2019 that contain free sugars (*n* = 67).

Food Categories and Sub-Groups	Stage 1 [4 m+]	Stage 2 [6 m+]	Stage 3 [8 m+]	Stage 4 [10 m+]	Stage 5 [12 m+]	Grand Total
**Breakfast**	**1**	**7**	**1**		**1**	**10**
Dry cereal	1	1				2
Ready-to-eat		6	1		1	8
**Desserts**		**15**	**6**			**21**
Custard & pudding		15	6			21
**Meals**	**9**	**2**	**3**		**3**	**17**
Fruit	7	2	2			11
Fruit & Vege	2					2
Meat			1		1	2
Poultry					1	1
Vegetables					1	1
**Snacks**			**4**	**2**	**13**	**19**
Rusk			1		2	3
Sweet			3	2	11	16
**Grand Total**	**10**	**24**	**14**	**2**	**17**	**67**

**Table 4 nutrients-12-02980-t004:** Types of vegetables used in commercially available complementary foods (CACFs) in New Zealand supermarkets in 2019 along with their total sugar content (g/100 g) and frequency of use (number and percentage).

Vegetable	Sugar Content (g/100 g)	Number and Percentage of CACFs Containing Vegetables
Carrot	6.6	54 (27.4%)
Onion	5.1	43 (21.8%)
Kumara	9.2	31 (15.7%)
Tomato	1.9	31 (15.7%)
Pumpkin	4.1	29 (14.8%)
Sweetcorn	7.6	26 (13.2%)
Potato	0.3	22 (11.2%)
Peas	5.8	21 (10.7%)
Spinach *	0	12 (6.1%)
Zucchini	0.6	10 (5.1%)
Celery	1.3	9 (4.6%)
Parsnip	5.9	7 (3.6%)
Courgette	0.6	5 (2.5%)
Red Capsicum	4.9	3 (1.5%)
Cauliflower *	1.8	3 (1.5%)
Broccoli *	1.8	3 (1.5%)
Green beans	3.2	2 (1.0%)
Leek	3.9	2 (1.0%)
Capsicum	2.8	2 (1.0%)
Kale *	0.5	2 (1.0%)
Squash	4.8	1 (0.5%)
Mushroom	0.6	1 (0.5%)
Avocado	0.5	1 (0.5%)
Cabbage *	3.1	1 (0.5%)

A relatively sweet vegetable (shaded) was one which contained more than 4 g/100 g of total sugar. Spinach and cruciferous vegetables are bitter vegetables which are demarcated with an asterisk (*).

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
