# Peer review of "Nutritional Aspects of Commercially Available Complementary Foods in New Zealand Supermarkets"

_nutrients, 2020, doi:10.3390/nu12102980_

Round 1
Reviewer 1 Report
Comments to authors: The topic of this paper is timely and of interest as the market and availability of commercial complementary foods is growing. The appropriateness of these foods for infants and young children needs to be considered. This paper needs to be more clear and specific about its objectives. It also does not assess labelling practices (presence/absence of nutritional content information, nutrient content claims, health claims, etc) and so the title needs to be revised or these aspects of labelling need to be added. Also, several key resources have been published around guidelines for appropriate CCFs by the WHO, which were not mentioned in this paper's introduction/methods and leave the reader wondering if the authors have a full grasp of the topic. A more comprehensive literature review on global guidance for CCFs and other studies that have been assessing nutritional content and labelling practices of these products is necessary. Finally, the paper needs to be properly edited. It was a challenge to review in parts due to grammar and language errors.
Lines 28-30: This is quite a long run-on sentence, could consider breaking into two.
Line 31: ‘a bridging period from six to twenty-four months of age’ is a time (not a place) so should be followed with ‘when a child learns….’ (not ‘where a child learns’)
Line 35: should read ‘which ARE mediated’
Line 38 – 39: ‘the use of CCFs HAS become…’
Line 39: ‘commonplace’ (delete ‘a’ before)
Lines 44-46: Here the tense goes back and forth between present and future
Line 50: the semi-colon is not appropriate punctuation
Line 52: An additional reference that must be added is the 2017 “WHO Guidance on ending the inappropriate promotion of foods for infants and young children”. This is global guidance for nations to adopt policies related to labeling and marketing. It was not referenced in this manuscript and so may be that the authors are not familiar with it. Given that the topic of this paper is labeling and packaging, this should absolutely be considered in the assessment of these NZ labels.
Line 56: CCFs which have increments in…. This is confusing, what is an ‘increment’ in complexity or variety or flavor?
Line 57: Again, not appropriate use of semi-colon
Lines 62-64: Global recommendations are to begin complementary feeding at 6 months, not before, because there is a risk for young child health and nutrition, particularly in low and middle income countries. Would recommend removing this caveat of the sentence as it is not universally valid.
Lines 79-81: packaging does not determine nutritional content (the ingredients determine nutritional content). Squeezable pouches need to have purees and these are often fruit-based hence the higher sugar content. This sentence is confusing, would suggest deleting or mentioning the inappropriate use of pouches marketed for children over 1 year of age, which the WHO guidelines highlight as children over 1 should begin consuming textures that requiring chewing.
Lines 76-87: Would recommend framing this paragraph around what the actual national guidelines are in New Zealand (or global recommendations) rather than citing previous research so that the background for this study is presented and it is clear to readers what you will be assessing/comparing against.
Lines 88-90: The specific objectives are not presented – what specific nutrient content is being summarized, what components of the national guidelines and food standards are being considered? Additionally, nothing is mentioned about labelling/packaging but this is in the paper title and mentioned in the introduction – is this also an objective?
Line 94: A nutrient profile is usually from nutrient profiling, which is the classification/ranking of foods based on their nutritional composition. I think instead the authors mean ‘label information’?
Line 98: the CCF acronym has already been noted above
Line 100: Infant formula is not defined as a complementary food by the WHO. It is defined as a breast-milk substitute (see International Code of Marketing of Breastmilk Substitutes).
Line 102: Why were products without age recommendations initially included? Did these have an image/language on the label that indicated they were appropriate for babies/toddlers? If so, these should be included as consumers will consider them as complementary foods.
Line 111: What is’ observational descriptive analysis’? Descriptive analysis is a statistical term – I believe the authors mean that the database was reviewed, and products categorized accordingly? Or if the packaging details were not in the database, did the authors go to shops and look at products and categorize?
Line 131 – 132: The presence of ANY added sugar was used to define a product as ‘sweet’? Is there a reference for this? Many savory products contain added, sugar. And what about products that are reconstituted with water? Added sugar in these products is absolutely a problem, but this seems a stretch as a definition of something that has a ‘sweet’ flavor without any quantification of the sugar content.
Lines 137 – 147: These selections of the guidelines should be introduced earlier, as they are what the analyses are based on.
Lines 142-143: These sodium limits are for infants less than 12 months. What was used in cases where products were ‘Stage 5’ (over 12 months)?
Lines 144-145: Texture is noted in the introduction and here as a bullet and in the results, but no information is provided in the methods around how this was assessed. Please provide details on how texture/variety/flavor/amount were defined and measured and assessed, as well as how the ‘increase’ or ‘gradient’ across age was evaluated.
Line 154: A definition of a ‘discrete CCF’ is needed – do different flavors of the same product count all as 1? Or are these each a separate product? And what about variation in packaging – does a single serving package of a product count as a separate product or not?
Line 160: grammatical error
Line 181: But this salt threshold is for children below 12 months – what is the appropriate threshold for products for older young children?
Lines 229 – 230: How were textures evaluated? Were there photos in the database? And how were the products analyzed to determine that they ‘became more complex’ with increasing stages? These details are missing entirely from the methods.
Lines 235 – 236: This is a result, and since it is beign presented for the first time in this paper, it needs to be in the results section. Also, is this based on all the criteria that were assessed? Were those assessed in this paper ALL the criteria in the guidelines? If not, please note that they did not pass the guidelines that were actually assessed in this study.
Lines 256 – 258: Why was this analysis not done? It easier could be. And it was also highlighted in the introduction. Please include this analysis.
Line 262: Recent guidance from WHO Europe also notes that squeezable pouches should not be marketing for children below 12 months for this reason. Please consider noting this in your discussion and also noting how many products were in squeezable pouches for children OVER 12 months. Ref: Ending inappropriate promotion of commercially available complementary foods for infants and young children between 6 and 36 months in Europe (2019)
Reviewer 2 Report
This paper is in accordance with similar ones published in the last few years studying the same topic in different markets, mainly in European countries. This research has its main interest in the national scenario, and it is a way to measure how food industry follows national recommendation when dealing with baby foods.
One concerns arise in my opinion:
- It is very unlike to use Commercial complementary foods as the unique way to provide complementary foods in infants and toddlers. If this is not the case, the authors should provide some data on the subject. In this sense the considerations regarding introduction of common food allergens, flavors and textures are probably unrealistic.
In any case, the paper is appropriately designed and developed and contains useful information for pediatricians, dietitians and families.
